# Healthcare providers experiences of using uterine balloon tamponade (UBT) devices for the treatment of post-partum haemorrhage: A meta-synthesis of qualitative studies

Kenneth Finlayson[1]☯*, Joshua P. Vogel[2]☯‡, Fernando Althabe[3]‡, Mariana Widmer[3]‡, Olufemi T. Oladapo[3]☯

1 Research in Childbirth and Health (ReaCH) Unit, University of Central Lancashire, Preston, Lancashire, United Kingdom, 2 Maternal, Child and Adolescent Health Program, Burnet Institute, Melbourne, Australia, 3 UNDP/UNFPA/UNICEF/WHO/World Bank Special Programme of Research, Development and Research Training in Human Reproduction (HRP), Department of Sexual and Reproductive Health and Research, World Health Organization, Geneva, Switzerland

☯ These authors contributed equally to this work.
‡ These authors also contributed equally to this work.
* kwfinlayson1@uclan.ac.uk

**Data Availability Statement:** All relevant data are within the manuscript and its Supporting Information files.

## Abstract

### Background

Postpartum haemorrhage (PPH) is a leading cause of maternal mortality and severe morbidity globally. When PPH cannot be controlled using standard medical treatments, uterine balloon tamponade (UBT) may be used to arrest bleeding. While UBT is used by healthcare providers in hospital settings internationally, their views and experiences have not been systematically explored. The aim of this review is to identify, appraise and synthesize available evidence about the views and experiences of healthcare providers using UBT to treat PPH.

### Methods

Using a pre-determined search strategy, we searched MEDLINE, CINAHL, PsycINFO, EMBASE, LILACS, AJOL, and reference lists of eligible studies published 1996–2019, reporting qualitative data on the views and experiences of health professionals using UBT to treat PPH. Author findings were extracted and synthesised using techniques derived from thematic synthesis and confidence in the findings was assessed using GRADE-CERQual.

### Results

Out of 89 studies we identified 5 that met our inclusion criteria. The studies were conducted in five low- and middle-income countries (LMICs) in Africa and reported on the use of simple UBT devices for the treatment of PPH. A variety of cadres (including midwives, medical officers and clinical officers) had experience with using UBTs and found them to be effective, convenient, easy to assemble and relatively inexpensive. Providers also suggested regular, hands-on training was necessary to maintain skills and highlighted the importance of community engagement in successful implementation.

**Funding:** The work was commissioned to the University of Central Lancashire by the UNDP/UNFPA/UNICEF/WHO/World Bank Special Programme of Research, Development and Research Training in Human Reproduction (HRP), a cosponsored program executed by the World Health Organization (WHO).

**Competing interests:** The authors have declared that no competing interests exist.

## Conclusions

Providers felt that administration of a simple UBT device offered a practical and cost-effective approach to the treatment of uncontrolled PPH, especially in contexts where uterotonics were ineffective or unavailable or where access to surgery was not possible. The findings are limited by the relatively small number of studies contributing to the review and further research in other contexts is required to address wider acceptability and feasibility issues.

## Introduction

Recent estimates of global maternal mortality rates show a decline of approximately 35% between 2000 and 2017 [1]. However, this still equates to approximately 300,000 maternal deaths in 2017, with more than 94% of deaths occurring in low-income countries [1]. More than a quarter of these deaths are due to haemorrhage, with postpartum haemorrhage (PPH) accounting for almost 20% of all direct causes of deaths [2].

In functional terms, a primary PPH is generally defined as a blood loss of 500 ml or more within 24 hours after birth [3]. However, in some settings where pregnant women are well nourished, unlikely to be anaemic, and where grand multiparity is uncommon, the blood loss threshold used to define PPH may be higher, such as over 1000 mL [4]. Conversely, for women who are malnourished or severely anaemic, postpartum blood loss of less than 500 mL may cause more severe long-term adverse consequences.

In a review of studies exploring PPH risk factors Oyelese and Ananth [5] indicate that labor induction, fetal macrosomia, retained placenta/prolonged third stage, prolonged labor, cesarean delivery, genital tract trauma and non-use of oxytocin or other uterotonic agents in the third stage are consistently associated with PPH across different populations [5]. However, individual risk factors are likely to be poor predictors of PPH occurrence [6] and the appropriate care practices to reduce the risk of PPH are not well defined or carried out effectively, particularly in low income settings [7,8].

The most common cause of PPH is uterine atony, a condition in which the uterus fails to contract after the birth of the baby [9]. Current WHO guidance on treatment for PPH due to uterine atony comprises two major elements—first response interventions, including uterotonic drugs (primarily oxytocin), isotonic crystalloids, tranexamic acid and uterine massage; and, if necessary, interventions for refractory PPH, including compressive measures (aortic compression or bimanual uterine compression), Intrauterine balloon tamponade and non-pneumatic anti shock garment (NASG) [10]. If these measures fail, surgical intervention (i.e. compressive sutures, artery ligation or hysterectomy) is required [11]. In 2012, the World Health Organization (WHO) made a conditional recommendation in favour of the use of uterine balloon tamponade (UBT) in situations where uterotonics were not available or were not effective in controlling bleeding [3].

While there are several different UBT devices, they essentially consist of a balloon, a catheter and a syringe. The UBT works by inserting the balloon into the uterus and inflating it (usually with sterile water)—the subsequent increase in intra-uterine pressure is expected to reduce blood flow and promote clotting in the uterine wall [11]. While a UBT can be improvised from readily available items (such as gloves, condoms and Foley catheters), purpose-designed UBT devices are commercially available (e.g. Sengstaken-Blakemore or Bakri Balloon) [12]. In low-income settings (particularly in rural locations where resources may be limited, and transfer to higher level facilities for invasive surgical intervention (e.g. removal of the uterus) may be

difficult, the use of a UBT has shown promise in the management of primary PPH [13,14]. However, the technique remains under-researched and under-utilized and issues pertaining to the feasibility and acceptability of using a UBT have not been fully evaluated [14]. With this in mind we set out to explore the views, perceptions and experiences of relevant stakeholders (including physicians, nurses, midwives and skilled birth attendants of using any form of UBT to treat PPH. Evidence from this study will be used to inform the updating of WHO recommendations on the treatment of PPH [15].

## Methods

We conducted a meta-synthesis of qualitative studies in accordance with the PRISMA guidelines (See S1 Table—PRISMA Checklist). We included studies where the focus was on healthcare providers use of a UBT (regardless of type of device) for the treatment of primary PPH. Study assessment included the use of a validated quality appraisal tool [16]. Thematic synthesis techniques [17] were used for analysis and synthesis, and GRADE-CERQual [18] was applied to the resulting review findings.

### Criteria for inclusion

The focus of the review was the experiences of healthcare professionals regarding their use of UBT for the treatment of PPH. We initially sought to include the views of women as well (relating to their experiences of receiving a UBT to treat PPH) but our searches failed to identify any relevant studies. We included qualitative studies reporting first-hand accounts (including the views, experiences and perceptions) of healthcare providers who used a UBT to treat a PPH in a health facility. We did not specify use of a particular type of UBT and included improvised devices as well as purpose-designed and commercially available devices. No language restriction was imposed. Case studies, conference abstracts and grey literature were not included.

### Search strategy

A search strategy was developed using a PEO (Population, Exposure, Outcome) structure with the addition of search limiters to identify qualitative studies. Systematic searches were carried out in December 2019 in Medline [OVID]; CINAHL [EBSCO]; PsycINFO [EBSCO]; EMBASE [OVID]; LILACS (for studies conducted in South America) and AJOL (for studies conducted in Africa). Searches were carried out using keywords for the Population, Exposure and Outcomes criteria and methodology where possible, or for smaller databases, using exposure keywords only. An example of the search strategy for OVID MEDLINE is shown in Box 1 below:-

Box 1 –Example of search strategy (Ovid MEDLINE)

Ovid MEDLINE(R) and Epub Ahead of Print, In-Process & Other Non-Indexed Citations and Daily 1946 to December 14[th] 2019

1. (uterine balloon tamponade or uterine balloon or uterine tamponade or UBT or intra-uterine tamponade or intrauterine balloon or balloon tamponade or hydrostatic tamponade or balloon dilatation or condom catheter or Bakri balloon or Sengstaken-Blakemore tube or Rusch balloon or Foley catheter).ti,ab.

2. (staff or provider* healthcare provider* or healthcare professional* or nurs* or midwife or midwives or physician* or doctor* or obstetrician* or clinician* or skilled birth attendant*).ti,ab.

3. (view* or experience* or expectation or perspective* or perception* or opinion* or belief* or understand* or encounter* or attitude* or prefer* or provision or feel* or think or thought* or value*).ti,ab.

4. 1 AND 2 AND 3

5. Qualitative Research/

6. (qualitative or interview* or focus group* or ethnograph* or phenomenol* or mixed methods or grounded theory).ti,ab.

7. 5 or 6

8. 4 AND 7

KF collated records into an Excel file, removed duplicates, and where possible, removed irrelevant records by title. KF and JV assessed the remaining records by abstract against the a priori inclusion/exclusion criteria and any records not meeting the inclusion criteria were discarded. Any disagreements regrading inclusion were mediated by OO. The full text records of the remaining studies were retrieved, screened and assessed for inclusion against the inclusion criteria by KF and JV with any disagreements on inclusion mediated by OO. The full texts of studies published in languages other than English were translated into English using freely available online software (Google Translate).

## Quality assessment

Included studies were appraised using an instrument developed by Walsh and Downe [15] and modified by Downe et al [19]. Studies were rated against 11 pre-defined criteria, and then allocated a score from A–D, where A represented a study with no, or few flaws, with high credibility, transferability, dependability and confirmability; B, a study with some flaws, unlikely to affect the credibility, transferability, dependability and/or confirmability of the study; C, a study with some flaws that may affect the credibility, transferability, dependability and/or confirmability of the study; and D, a study with significant flaws that are very likely to affect the credibility, transferability, dependability and/or confirmability of the study. Studies scoring C or higher were included in the final analysis. KF conducted the quality assessment of all included papers and a final grade was agreed by consensus between all review authors.

## Data extraction and analysis

The analytic process broadly followed the principles of thematic synthesis [17]. We started by reading the papers closely and identified an index paper that best reflected the focus of the review. The themes and findings identified by the authors of the index paper were coded and entered onto a spreadsheet by KF to develop an initial thematic framework. The findings of the remaining papers were coded and mapped to this framework, which continued to develop as the data from each paper were added. This process includes looking for what is similar between papers and for what contradicts ('disconfirms') the emerging themes. For the disconfirming process all authors consciously looked for data that would contradict our emerging

themes, or our prior beliefs and views related to the topic of the review. Data extraction and synthesis proceeded concurrently. Descriptive themes were developed from the quote material and author interpretations of the studies.

Once the framework of descriptive review findings was agreed by all of the authors, the level of confidence in each review finding was assessed using the GRADE-CERQual tool [18]. GRADE-CERQual assesses the methodological limitations and relevance to the review of the studies contributing to a review finding, the coherence of the review finding, and the adequacy of data supporting a review finding. Based on these criteria, review findings were graded for confidence using a classification system ranging from 'high' to 'moderate' to 'low' to 'very low'. The data extraction, analysis, synthesis and CERQual grading stages are illustrated in the S1 Appendix. Following CERQual assessment the review findings were grouped into higher order, analytical themes and the final framework was agreed by consensus amongst the authors.

## Results

### Included studies

Our electronic searches yielded 122 citations. We screened 89 unique records after duplicate removal and excluded 68 following title and abstract screening. 21 full-text articles were assessed for eligibility and five studies were included in the qualitative synthesis [20–24]. A PRISMA flow chart illustrating the selection process is shown in Fig 1.

The five studies all took place in low or lower-middle income countries in Africa- 2 in Kenya; 1 each in Sierra Leone and South Sudan and 1 across two countries (Kenya and Senegal). They were conducted between 2013 and 2016 and included the views of more than 200 participants. The participants worked in a range of health facilities, including community-based rural health centres and urban hospitals. All of the studies were conducted by the same or similar review team reporting on provider experiences of UBT devices. Three of the studies [20,21,24] were undertaken following a comprehensive PPH training package called, 'Every Second Matters for Mothers and Babies–Uterine Balloon Tamponade' (ESM-UBT), in which facility based healthcare staff attended a 3 hour workshop where they were taught how to use to use a UBT within the context of the established national protocols for PPH management. The UBT used in the training programme was a relatively basic device consisting of a syringe, a valve and a condom fastened to the end of a Foley catheter with cotton string. One study explored perceptions of using UBT for uncontrolled PPH following a similar ESM-UBT training programme in community settings in South Sudan [23] and the other explored the use of improvised UBT devices (including condoms and surgical gloves, plus catheters, syringes and string) amongst staff at healthcare facilities in Kenya [22]. They represent the views of a variety of professional disciplines including obstetricians, midwives, nurse-midwives and medical officers.

All of the studies were qualitative and descriptive in design and ranged in quality from B to C+. Further details of the included studies are shown in Table 1.

## Findings

Six descriptive themes (review findings) were derived and assessed for confidence using the CERQual tool (see Table 2). Three were rated as moderate, two low and one very low (See S1 Appendix for details of study assessment, data extraction and CERQual Gradings). These descriptive themes were synthesised into four interpretive themes, Life-saving support; Affordable Benefits, Practically Simple and Community Awareness, described below:

## Life-saving support

Providers generally viewed the UBT as an effective treatment option for PPH in situations where uterotonics were either unavailable or ineffective. Health providers felt that a basic UBT

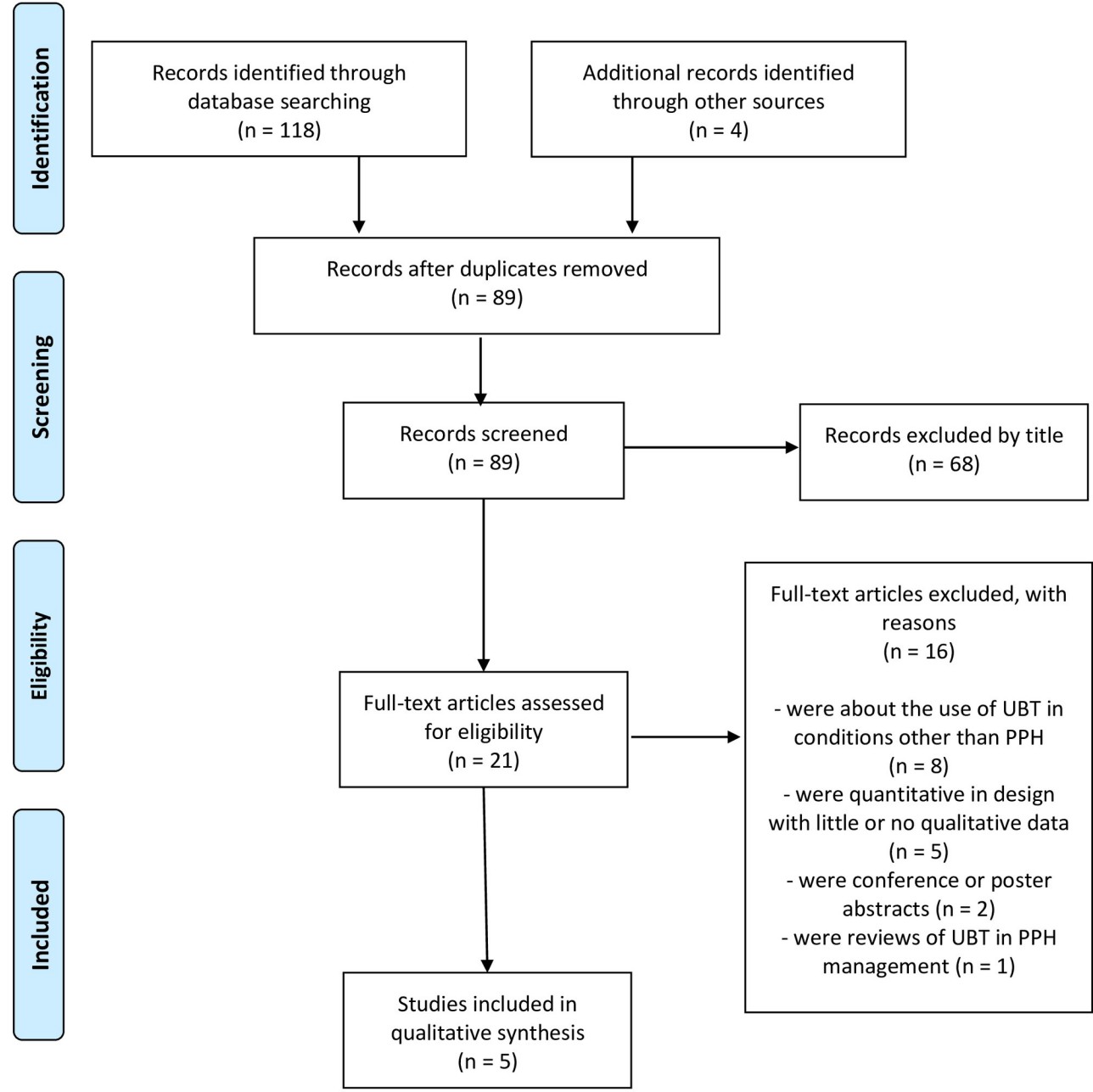

*From:* Moher D, Liberati A, Tetzlaff J, Altman DG, The PRISMA Group (2009). *P*referred *R*eporting *I*tems for *S*ystematic Reviews and *M*eta-*A*nalyses: The PRISMA Statement. PLoS Med 6(6): e1000097. doi:10.1371/journal.pmed1000097

**For more information, visit www.prisma-statement.org.**

**Fig 1. PRISMA flowchart goes here.**

**Table 1. Study characteristics.**

| Study Number | Authors and Date | Country | Resource | Participants | Context | Design and Method | Data Collection | UBT design | Quality Rating |
|---|---|---|---|---|---|---|---|---|---|
| 1 | Natarajan et al, 2015 [20] | Kenya | Lower Middle | Nurse-Midwives, Clinical Officers, Medical Officers | Urban & Rural | Descriptive, qualitative informed by interviews | 68 semi-structured interviews relating to 31 cases of UBT use | Condom-catheter UBT | C+ |
| 2 | Natarajan et al, 2016a [21] | Kenya | Lower Middle | Obstetricians, Nurse-Midwives, Medical Officers, Clinical Officer | Urban & Rural | Descriptive, qualitative informed by interviews | 29 semi-structured interviews relating to 29 cases of improvised UBT use | Improvised UBT | C+ |
| 3 | Natarajan et al, 2016b [22] | Sierra Leone | Low | Doctors, Midwives, Nurses, Clinical Officers | Urban | Descriptive, qualitative informed by interviews | 61 semi-structured interviews relating to 30 cases of UBT use | Condom-catheter UBT | C+ |
| 4 | Nelson et al, 2013 [23] | South Sudan | Low | Community-based Birth Attendants, Facility-based Providers, Patients, Patient relatives and other community members | Largely Rural | Descriptive, qualitative informed by interviews | 24 semi-structured interviews relating to 13 cases of UBT use | Condom-catheter UBT | B |
| 5 | Pendleton et al, 2016 [24] | Kenya & Senegal | Lower Middle | Doctors | Urban & Rural | Descriptive, qualitative informed by interviews | 30 semi-structured interviews relating to 80 cases of UBT use | Condom-catheter UBT | B- |

device prevented uncontrolled bleeding in women experiencing a primary PPH. The device appeared to be effective in patients with various levels of clinical severity, from those who were clinically stable to those who displayed signs of severe shock.

> *"In all of my three cases, they were all success stories. I put in my UBT and then like magic I observed. . . in the next 10 min, she was completely dry. No bleeding at all".- Ob/Gyn resident, [20]*

Although providers had minimal experience with UBT's, largely because of the effectiveness of uterotonics and because only a small proportion of women experience refractory PPH, they frequently discussed the life-saving capacity of the device in situations where there were limited options and limited time.

> *"We do transfer. But it's very far. And getting the ambulance here the patient can die. And there are some seasons the ambulance can't come because it's muddy. So the UBT is very important and should be there in place in the rural hospitals in plenty".—Nurse [21]*

## Affordable benefits

Providers referred to the cost of the device directly (its relatively cheap price) as well as indirectly, in terms of the potential cost-savings associated with UBT use. All of the health providers contributing to this review reported on their experience of using a basic UBT costing less than $5.00 per device. Providers were acutely aware of the resource constraints in the settings where they worked and recognized the importance of using an inexpensive, potentially life-saving treatment:-

> *"In the resource limited setups where I used to work, where you don't have all the things that you need to do all of the complex procedures, a simple thing like the UBT kit done effectively would ensure that you save a life".—Medical officer, [22]*

**Table 2. Summary of findings.**

| Descriptive theme (Review finding) | Studies contributing to the review finding | CERQual grading | Analytical theme |
|---|---|---|---|
| **Perception that UBT is highly effective**—Healthcare providers believed that UBT was highly effective in arresting the blood loss associated with PPH. Following the use of uterotonics, UBT was seen as a valuable secondary option(following treatment with uterotonics) with frequent positive outcomes appreciated by both health professionals and local communities. | 5 studies:- Natarajan et al (2016) Kenya; Natarajan et al (2015) Kenya; Natarajan et al (2016) Sierra Leone; Nelson et al (2013), South Sudan; Pendleton et al (2016); Kenya & Senegal | Moderate | Life-Saving Support |
| **Perception that UBT reduces the need for surgery or referral**—In contexts where UBT had been introduced healthcare providers highlighted a reduction in the use of surgery (hysterectomy) to control blood loss and/or referral to secondary or tertiary level facilities for surgical procedures. The benefits associated with women not having surgery, in terms of the psychological impact, the loss of fertility and the additional costs or difficulties associated with transport were also identified by some providers. | 5 studies:- Natarajan et al (2016) Kenya; Natarajan et al (2015) Kenya; Natarajan et al (2016) Sierra Leone; Nelson et al (2013), South Sudan; Pendleton et al (2016); Kenya & Senegal | Moderate | Affordable Benefits |
| **Practical and affordable solution to PPH treatment**—UBT was seen as a practical and cost-effective solution to PPH treatment in areas where resources may be limited. Healthcare providers in one location highlighted the widespread availability of items (condoms, surgical gloves) that could be improvised into an effective balloon tamponade, although some providers discussed the lack of formal UBT devices in health facilities. | 4 studies:- Natarajan et al (2016) Kenya; Natarajan et al (2016) Sierra Leone; Nelson et al (2013), South Sudan; Pendleton et al (2016); Kenya & Senegal | Low | |
| **Perception that UBT is easy to use**—Healthcare providers believed that use of the UBT required minimal expertise with the added advantage that the technique could be taught to lower level cadres relatively easily. Some providers felt that the 'ease of use' was particularly advantageous in relatively remote settings where resources may be limited. | 5 studies:- Natarajan et al (2016) Kenya; Natarajan et al (2015) Kenya; Natarajan et al (2016) Sierra Leone; Nelson et al (2013), South Sudan; Pendleton et al (2016); Kenya & Senegal | Moderate | Practically Simple |
| **Desire for more training**—Despite the perception that the UBT was relatively easy to use healthcare providers felt that they needed regular 'hands on' training to gain experience with the technique. Some providers also felt unsure about how long to leave a UBT in place and stressed the importance of having easy access to a UBT during the chaotic and often stressful management of a PPH | 4 studies:- Natarajan et al (2016) Kenya; Natarajan et al (2015) Kenya; Natarajan et al (2016) Sierra Leone; Nelson et al (2013), South Sudan | Low | |
| **Appreciation of cultural beliefs**—Some providers were aware of a reluctance amongst some women to use the device because of fears of infertility. A few providers also felt that some women were reluctant to have a condom inserted because of religious or cultural beliefs and suggested the 'device' should be referred to as a tamponade rather than a condom to allay these concerns. | 2 studies:- Natarajan et al (2015) Kenya; Nelson et al (2013), South Sudan | Moderate | Community Awareness |

Providers also discussed the indirect, cost-saving benefits relating to reduced referral rates and/or the impact on surgical interventions. Many of the providers contributing to this review worked in relatively remote locations where referral to higher-level facilities could be challenging. Under these circumstances, and given the immediacy of the situation, health providers felt that successful use of a basic UBT minimized referrals and reduced the need for potentially costly and dangerous journeys to higher level facilities. Furthermore, providers suggested that the use of a basic UBT reduced the requirement for expensive, surgical interventions that are prone to complications, particularly hysterectomy. The benefits of fewer surgical procedures were discussed in terms of cost savings as well as reduced levels of physical and psychological trauma for women.

"*The first thing is finance—the low cost as compared to hysterectomy. You don't need to give drugs, like surgery you need anaesthesia and so and so. You can go faster compared to surgery*".—Ob/Gyn resident, [20]

## Practically simple

In situations where a basic UBT was used many providers commented on their ease of use. In most circumstances' providers utilized a device consisting of a condom, syringe, catheter and valve but sometimes had to resort to making improvised devices using readily available items like rubber gloves instead of condoms and pieces of string instead of valves. The availability of device components and the ease of assembly encouraged a variety of cadres to treat women with a PPH. In addition to obstetricians the UBT device was also used successfully by mid-wives, nurse midwives, medical officers and clinical officers.

"*Eighteen (75%) of the 24 providers who inserted a UBT device reported no challenges during the process and frequently described the uterine balloon as "simple" and "easy to use". (Author Quote),* [21]

Although the vast majority of providers found the device easy to use some raised questions about when to insert the balloon and how long to leave it in place, whilst others recognized the need for 'hands-on' refresher courses given the relative infrequency of UBT use.

"*[The uterine balloon] is good, and also it need even to be trained again and again and give them encouragement so that even though they don't come across those patients [with PPH regularly], they will not forget".* [23]

## Community awareness

Providers occasionally discussed patient and community perceptions of UBT and although this was generally in positive terms there were a few issues relating to understanding and culture. Providers reported that some women were initially anxious or fearful about the insertion of a UBT because of the impact it might have on fertility:-

"*They were thinking that the balloons will block the uterus, and inside the womb, the mother will not bear any child's anymore. It will block".* [23].

Some providers also discussed using a condom in this way, particularly with communities who might be sensitive about the general use of condoms.

"*Further, three providers discussed their experiences with patients or family members—particularly in Garissa County, a conservative Islamic region of Kenya—who were uncomfortable being treated with a device that used a condom"* (Author Quote) [20]

In these situations, providers recognized the importance of community engagement prior to the introduction of UBT into local health services.

## Discussion

We set out to explore provider perceptions of UBT use for the treatment of PPH. Our theme of 'life saving support' suggests that the use of a relatively basic UBT may be of particular benefit in low-resource settings where access to more complex treatment options may be limited or challenging. Providers generally viewed the UBT as an effective, second-line solution to PPH management, irrespective of severity and clinical condition. Although the vast majority of PPH cases in the included studies were controlled with uterotonics (primarily oxytocin),

providers seemed impressed and reassured by the capacity of a relatively simple device to arrest blood loss. Their views would suggest that basic UBT devices may be particularly useful in contexts where uterotonics are unavailable because of supply chain problems [25] or, in the case of oxytocin, ineffectual because of heat sensitivity [26].

In addition, our theme relating to affordable benefit suggests that the minimal cost and relative availability of UBT components (condom, catheter and syringe) is likely to be an important consideration in low-resource contexts where health budgets may be stretched. The cost related benefits of using a basic UBT are further illustrated in the reduced rates of relatively expensive surgical procedures as well as dangerous and potentially costly transfers to higher-level facilities. Our findings also indicate that cost-savings may be generated by training lower level cadres to assemble and use UBTs in low-resource settings, especially in community facilities where more experienced health professionals may be in short supply [27,28]. A recent review exploring task shifting in active management of the third stage of labour (AMTSL) showed that, with minimal training, administration of uterotonics (primarily misoprostol) by unskilled health workers was acceptable, feasible and, importantly, reduced rates of PPH [29]. Similarly, findings from this review indicate that midwives, medical officers and clinical officers could be trained to administer a UBT safely and effectively.

Although most providers felt that training in UBT use was straightforward and relatively simple, the requirement for 'refresher' training was highlighted. Providers had queries about how to assemble a UBT and how long to leave it in place, but most of their requests related to further 'hands-on' training to gain confidence in the placement and insertion of the device. The issue of additional or refresher training in LMICs is a recurrent theme in qualitative reviews of maternity care practice amongst providers [30,31] and, although it remains unclear how to resolve the situation, it is worth highlighting again in this review for policymakers and other stakeholders who may be considering UBT implementation programmes.

Our theme relating to community awareness highlights the importance of community engagement in healthcare decision making. Previous qualitative reviews exploring women's experiences of maternity care indicate that an awareness of cultural and traditional practices is likely to lead to increased acceptability and compliance, particularly if the healthcare condition and/or related intervention is explained to the local population in a language and format that is engaging and understandable [30,32]. In this review, providers indicated that some women were initially fearful of using a UBT because of cultural understandings related to fertility, whilst other communities had misgivings about having a condom inserted because of sexually related religious beliefs.

If the Sustainable Development Goal (SDG 3) of reducing the maternal mortality ratio to 70 per 100,000 live births by 2030 is to be achieved [33] then simple, effective and inexpensive interventions to manage PPH must be available for women giving birth, particularly in limited-resource settings. Overall, our findings suggest that healthcare provider experiences with relatively basic or improvised UBTs were positive. However, it is important to note that UBTs are a rescue intervention for refractory PPH, and currently trial-based evidence relating to UBT efficacy and effectiveness remains unclear [34].

The absence of other important stakeholder input, especially from women, is a significant limitation of this study. We initially set out to include women's views of UBT but our preliminary scoping searches did not identify any studies. Given the potential impact on women and the intrusive nature of the intervention it is important that qualitative studies addressing women's experiences with UBT are conducted in a variety of settings and contexts. The provider themes discussed here should also be interpreted with caution because of the relatively small number of included studies and the largely descriptive nature of the findings in the primary papers. In addition, the studies were all conducted in low-and-middle-income countries

(LMICs) in Africa and were assessed as being of average quality. Arguably, we may have retrieved more studies if we had utilized language specific search terms in our search strategy but time constraints and the availability of researchers with appropriate language skills limited our capacity to do so. One of the notable features of this synthesis is the similarity of the included studies given that they were all conducted by the same or similar review team reporting on provider experiences of UBT following a comprehensive PPH training package called, 'Every Second Matters for Mothers and Babies–Uterine Balloon Tamponade' (ESM-UBT). This homogeneity rather limits the impact and transferability of the findings to other settings where different types of UBT used in different circumstances may yield additional insights.

## Conclusion

Based on a relatively small number of primary studies from low resource countries, the findings from healthcare providers indicate that use of a relatively simple UBT device represents a practical and inexpensive approach to the treatment of uncontrolled PPH. Providers in these settings suggest that a simple UBT device is accessible, relatively easy to use and may be a life-saving solution in situations where uterotonics are ineffective or unavailable or where access to surgery is not possible. These views would likely facilitate implementation of UBT, should rigorous research in low-resource settings indicate that UBT was effective in preventing adverse maternal outcomes related to PPH. However, the limited amount of data supporting this review suggests that more research is needed to assess acceptability, feasibility and implementation concerns across a range of different settings. Additional research is also needed in relevant communities to understand the views and experiences of women relating to the use of condom-catheter and/or improvised UBTs.

## Supporting information

**S1 Table. PRISMA checklist.**
(DOCX)

**S1 Appendix. Data extraction, analysis, synthesis and CERQual grading.**
(XLSX)

## Acknowledgments

We would like to thank all members of the WHO technical working group for their support and advice during the development of this review. This paper reflects the views of the named authors only, and not the views of their institutions. The authors declare they have no conflicts of interest.

## Author Contributions

**Conceptualization:** Kenneth Finlayson, Joshua P. Vogel, Fernando Althabe, Mariana Widmer, Olufemi T. Oladapo.

**Data curation:** Kenneth Finlayson.

**Formal analysis:** Kenneth Finlayson.

**Funding acquisition:** Olufemi T. Oladapo.

**Investigation:** Kenneth Finlayson.

**Methodology:** Kenneth Finlayson, Joshua P. Vogel, Fernando Althabe, Olufemi T. Oladapo.

**Project administration:** Kenneth Finlayson, Mariana Widmer.

**Resources:** Kenneth Finlayson.

**Supervision:** Joshua P. Vogel, Fernando Althabe, Olufemi T. Oladapo.

**Writing – original draft:** Kenneth Finlayson, Joshua P. Vogel, Fernando Althabe, Mariana Widmer, Olufemi T. Oladapo.

**Writing – review & editing:** Kenneth Finlayson, Joshua P. Vogel, Fernando Althabe, Mariana Widmer, Olufemi T. Oladapo.

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
