## [Decision Letter · Decision Letter 0]

22 Dec 2020

PONE-D-20-30548

Healthcare providers experiences of using Uterine Balloon Tamponade (UBT) devices for the treatment of post-partum haemorrhage (PPH): a meta-synthesis of qualitative studies

PLOS ONE

Dear Dr. Finlayson,

Thank you for submitting your manuscript to PLOS ONE. After careful consideration, we feel that it has merit but does not fully meet PLOS ONE’s publication criteria as it currently stands. Therefore, we invite you to submit a revised version of the manuscript that addresses the points raised during the review process.

The manuscript and the reviewers’ comments were carefully evaluated. The manuscript was appreciated by the Reviewers. Nevertheless, as suggested, the manuscript requires some improvement before to be considered for publication. Suggested revisions are in detail reported in the Reviewers’ comments.

We look forward to receiving your revised manuscript.

Kind regards,

Simone Garzon

Academic Editor

PLOS ONE

Journal Requirements:

2. Please provide any updates you might have since the original search was performed in XXX, or please provide the rational for ending your search at that time.

3.Please amend either the title on the online submission form (via Edit Submission) or the title in the manuscript so that they are identical.

Reviewers' comments:

Reviewer's Responses to Questions

**Comments to the Author**

1. Is the manuscript technically sound, and do the data support the conclusions?

Reviewer #1: Yes

Reviewer #2: Yes

Reviewer #3: Yes

2. Has the statistical analysis been performed appropriately and rigorously? 

Reviewer #1: N/A

Reviewer #2: Yes

Reviewer #3: N/A

3. Have the authors made all data underlying the findings in their manuscript fully available?

Reviewer #1: Yes

Reviewer #2: Yes

Reviewer #3: Yes

4. Is the manuscript presented in an intelligible fashion and written in standard English?

Reviewer #1: Yes

Reviewer #2: Yes

Reviewer #3: Yes

5. Review Comments to the Author

Reviewer #1: The writing quality is superb and style easy to read. The Methods are clear and of precise high quality. It is indeed unfortunate that more articles did not meet criteria for review.

It might be of interest for the authors to learn that in a difference-in-difference analysis of UBT implementation in Central India (to be published soon), the introduction of a UBT package for managing severe PPH was associated with a statistically significant reduction in the number of cases of a composite outcome of maternal adverse outcomes (death, hysterectomy or other surgeries)

Reviewer #2: Thank you for the opportunity to review this metasynthesis of qualitative studies on provider's experiences with UBT. This is a well-written and well-organized manuscript describing somewhat limited qualitative data. Below please find some line-item edits that I would recommend. In addition, the conclusion needs some work to highlight the limitations of the study and the available research and to provide implications of this study.

Abstract: include which methods you used to guide the qualitative metasynthesis.

Conclusions seem to be an overextension based on the methods and results described. If the question is what were providers' experiences, the conclusion should describe what providers' experiences were with UBT and then suggest some the clinical and future research implications of those findings.

Line 71: What are the statistics to support this statement?

Lines 77-78: List the first line treatments for uterine atony in order of use.

Lines 83-84: This recommendation isn't clear. Is this saying that the evidence is weak in support of using UBT or that the evidence comparing UBT to other interventions is weak?

Lines 96-97: This should be a statement of purpose or a statement of the primary question very explicitly stated. Make sure that the question/purpose is appropriate to the methods.

Table 2: The level of evidence doesn't match the text. The table appears to say 5 themes have moderate evidence and 1 has low but the text says 3 themes have moderate, 2 low, and 1 very low evidence.

Line 319: The homogeneity of the research team needs to be discussed much earlier in the paper. I think it's important to include in the results or discussion that these studies were conducted to evaluate the intervention program and how that influences the data (how people were trained and supported to use UBTs, etc).

Add the use of improvised UBTs to the discussion and conclusion sections. Is there evidence that these are effective, is it necessary or preferred to use specific devices or are the improvised ones just as good, etc.

Include a more in-depth discussion of the clinical, policy and future research implications. Where are there still gaps? Where is there sufficient evidence to make policy changes?

Reviewer #3: -Kudos for taking up a worthy cause. Highlighting the much needed evidence that supports such a life saving intervention at the WHO is indeed commendable however, uterine balloon tamponade is a well recognised PPH treatment modality and it's indication as not the first nor the last intervention might place it in a sort of catch-22 position: too few women require it compared to uterotonics however by the time more aggressive treatment is needed, providers may be opting to err on the side of caution by over-treating (surgical interventions) rather than taking the middle ground. If there really is a dearth of literature supporting UBT use then this may be why. On a related note, perhaps the inclusion criteria used were more likely to rule out relevant works rather than rule them in given that only 5 articles out of 21 made the final cut. These 21 being shortlisted from an initial 89. It's possible that the same complex interplay of socio-economic factors that maintains the disparity of PPH incidence and deaths between higher and lower resourced settings is responsible for the apparent lack of robust studies into an intervention that is rarely indicated at best.

-You rightly indicated that a major limitation to the of this meta-analysis is the fact that all the studies that 'made the cut' were conducted by the same team investigating a PPH management package that includes a condom balloon. This is another reason to take a second look at the inclusion criteria. Perhaps the data we seek is buried in the discussion of other UBT/PPH related works. Are there any older papers on the utilisation of tamponade in PPH perhaps when the Sengstaken-Blakemore was first introduced? I believe our focus and interest on the perceptions of healthcare providers is a relatively modern one and perhaps it's unfair for us to measure a relatively old intervention by it.

-While utilising Google Translate for a cursory search is efficient and practical, it may be advisable, given the epidemiology of

PPH to cast a wider net and conduct targeted searches (ideally by RAs proficient in)in other languages such as French, Arabic etc

-Line 90: typo in Sengstaken-Blake(r)more

-Line 133: Is "nurs" a typo?

6. PLOS authors have the option to publish the peer review history of their article (what does this mean?). If published, this will include your full peer review and any attached files.

Reviewer #1: **Yes: **Thomas F. Burke, MD

Reviewer #2: No

Reviewer #3: No

---

## [Author Response · Author response to Decision Letter 0]

10 Feb 2021

Thank you for considering our manuscript for publication. We appreciate your feedback and welcome the opportunity to address some of the reviewer’s concerns. 

We have copied & pasted the reviewer’s comments into this letter (see below) and will address each of them in turn. Our revised submission includes a copy of the revised manuscript with amendments (based on reviewer comments) in red font, as well as a clean copy of the manuscript incorporating all of the changes. 

We believe our manuscript meets with journal requirements but are happy to accept further guidance if this is not the case 

2. Please provide any updates you might have since the original search was performed in XXX, or please provide the rational for ending your search at that time.

We have run additional searches to ensure the review is up to date 

3.Please amend either the title on the online submission form (via Edit Submission) or the title in the manuscript so that they are identical.

The title has been amended on the online submission form

Reviewers' comments:

Reviewer's Responses to Questions

Comments to the Author

1. Is the manuscript technically sound, and do the data support the conclusions?

Reviewer #1: Yes

Reviewer #2: Yes

Reviewer #3: Yes

2. Has the statistical analysis been performed appropriately and rigorously? 

Reviewer #1: N/A

Reviewer #2: Yes

Reviewer #3: N/A

3. Have the authors made all data underlying the findings in their manuscript fully available?

Reviewer #1: Yes

Reviewer #2: Yes

Reviewer #3: Yes

4. Is the manuscript presented in an intelligible fashion and written in standard English?

Reviewer #1: Yes

Reviewer #2: Yes

Reviewer #3: Yes

5. Review Comments to the Author

Reviewer #1: The writing quality is superb and style easy to read. The Methods are clear and of precise high quality. It is indeed unfortunate that more articles did not meet criteria for review.

Thank you for these comments

It might be of interest for the authors to learn that in a difference-in-difference analysis of UBT implementation in Central India (to be published soon), the introduction of a UBT package for managing severe PPH was associated with a statistically significant reduction in the number of cases of a composite outcome of maternal adverse outcomes (death, hysterectomy or other surgeries).

Thank you. This is useful information but unfortunately we can’t include the study if we can’t reference it 

Reviewer #2: Thank you for the opportunity to review this metasynthesis of qualitative studies on provider's experiences with UBT. This is a well-written and well-organized manuscript describing somewhat limited qualitative data. Below please find some line-item edits that I would recommend. In addition, the conclusion needs some work to highlight the limitations of the study and the available research and to provide implications of this study.

Thank you for these comments. We have addressed specific items below

Abstract: include which methods you used to guide the qualitative metasynthesis.

We refer to ‘thematic synthesis’ (Thomas & Harden, 2008) in the abstract section (under methods) as the meta-synthesis method and also indicate that we used GRADE-CERQual to assess confidence in the findings. We trust this is sufficient as we are limited by word count in the abstract but further details of the methodology are outlined in the text. 

Conclusions seem to be an overextension based on the methods and results described. If the question is what were providers' experiences, the conclusion should describe what providers' experiences were with UBT and then suggest some the clinical and future research implications of those findings.

We have reworded the conclusion in both the abstract and the manuscript to summarize the findings and have added a sentence to reflect the limited number of studies in the review and the subsequent potential for further research in this area. The experiences of the providers are largely highlighted in the Findings section of the abstract. 

Line 71: What are the statistics to support this statement?

This evidence comes from a comprehensive review of PPH exploring epidemiology, trends and rates across different population groups and the results are not presented in straightforward statistical terms. However, we have amended this sentence to better reflect the authors findings.

Lines 77-78: List the first line treatments for uterine atony in order of use.

We have reworded these sentences to align with the care bundle approach for the treatment of PPH due to uterine atony recently recommended by the WHO (see ref 10). This document specifies an order for first-line treatment as well as an order for refractory PPH. These are now listed in the text. 

Lines 83-84: This recommendation isn't clear. Is this saying that the evidence is weak in support of using UBT or that the evidence comparing UBT to other interventions is weak?

Sorry this recommendation isn’t clear. The WHO issues either ‘strong’ or ‘weak’ (conditional) recommendations based on a number of indicators relating to the quality and robustness of the evidence (both quantitative and qualitative) supporting the use of a particular intervention for a specific condition. In 2012, the recommendation relating to the use of UBT for the treatment of PPH was supported by low quality evidence so the recommendation became conditional i.e. to be used only in situations where uterotonics were not available or not effective in controlling bleeding. We have removed the word ‘weak’ from the text to avoid any confusion. 

Lines 96-97: This should be a statement of purpose or a statement of the primary question very explicitly stated. Make sure that the question/purpose is appropriate to the methods.

Thank you. We have added some more detail to this sentence to make it more explicit. 

Table 2: The level of evidence doesn't match the text. The table appears to say 5 themes have moderate evidence and 1 has low but the text says 3 themes have moderate, 2 low, and 1 very low evidence.

Thank you for pointing this out. The text in the manuscript is correct and we have altered the table to align with the text.

Line 319: The homogeneity of the research team needs to be discussed much earlier in the paper. I think it's important to include in the results or discussion that these studies were conducted to evaluate the intervention program and how that influences the data (how people were trained and supported to use UBTs, etc).

Thank you for this observation. We agree that it makes sense to introduce the homogeneity of the research team at a much earlier stage and have now highlighted this issue in the first paragraph of the Results section. We have also added a few sentences immediately after this point outlining the training and type of UBT used in the intervention studies [refs 20,21,23,24] as well as the improvised UBT devices used in the remaining study [22]. 

Add the use of improvised UBTs to the discussion and conclusion sections. Is there evidence that these are effective, is it necessary or preferred to use specific devices or are the improvised ones just as good, etc.

Thank you for raising these queries. We have now described all of the UBT devices used in the included studies in the first section of the Results section (see comment above relating to line 319). The four intervention studies used a relatively simple, inexpensive (<$5.00) UBT kit consisting of a syringe, valve and a condom fastened to the end of a Foley catheter with cotton string. The remaining study looked at improvised devices using the same or similar items described in the kit. None of the studies utilized more complicated, expensive or patented devices so we refer to basic, inexpensive UBT devices throughout the narrative to reflect the composition of the devices outlined in all of the included studies. We believe our first descriptive theme highlighting the effectiveness of these simple UBT devices along with our interpretive theme, ‘Life-saving Support’, makes it clear that all of the UBT devices (including improvised adaptations) were perceived to be effective by healthcare providers. Our discussion and conclusion sections refer to these simple UBT devices and we don’t distinguish between those supplied in the intervention kits and the improvised devices highlighted in study ref 22 for the reasons outlined above. We trust this response answers your queries but if you’d like us to add something more specific then we’d be happy to consider it. 

Include a more in-depth discussion of the clinical, policy and future research implications. Where are there still gaps? Where is there sufficient evidence to make policy changes?

We have added another paragraph to the discussion incorporating two additional references. The paragraph further outlines some of the implications of our review and how it fits within the wider literature on UBT for PPH. We feel that our narrative already includes information relating to further research and research gaps – the discussion makes reference to the lack of data from women as a notable research gap [lines 344-348] as well as limited data on the different types of UBT and we also recommend further research amongst providers in a wider range of settings (given the homogeneity of the data in this review). We have also added another sentence to the conclusion to draw attention to these recommendations. With regard to policy changes we indicate in the introduction (final sentence) that the findings from this review will be used to inform the upcoming WHO guidelines on the treatment of PPH. However, given the limited amount of data in this synthesis it is unlikely to have a significant impact on global policy relating to the use of UBT for PPH treatment. Hence the recommendations for further research. 

Reviewer #3: -Kudos for taking up a worthy cause. Highlighting the much needed evidence that supports such a life saving intervention at the WHO is indeed commendable however, uterine balloon tamponade is a well recognised PPH treatment modality and it's indication as not the first nor the last intervention might place it in a sort of catch-22 position: too few women require it compared to uterotonics however by the time more aggressive treatment is needed, providers may be opting to err on the side of caution by over-treating (surgical interventions) rather than taking the middle ground. If there really is a dearth of literature supporting UBT use then this may be why. On a related note, perhaps the inclusion criteria used were more likely to rule out relevant works rather than rule them in given that only 5 articles out of 21 made the final cut. These 21 being shortlisted from an initial 89. It's possible that the same complex interplay of socio-economic factors that maintains the disparity of PPH incidence and deaths between higher and lower resourced settings is responsible for the apparent lack of robust studies into an intervention that is rarely indicated at best.

Thank you for these observations. We are aware of more quantitative and survey based studies exploring the use of UBT in different contexts, largely in LMICs in Africa. However, these too are limited in number and are rarely supported by qualitative data. We largely agree with your comments relating to socio-economic disparities and PPH incidence and hope that this review stimulates further discussion and, ultimately, a desire to conduct well designed studies (both quantitative and qualitative) in appropriate LMIC contexts. 

-You rightly indicated that a major limitation to the of this meta-analysis is the fact that all the studies that 'made the cut' were conducted by the same team investigating a PPH management package that includes a condom balloon. This is another reason to take a second look at the inclusion criteria. Perhaps the data we seek is buried in the discussion of other UBT/PPH related works. Are there any older papers on the utilisation of tamponade in PPH perhaps when the Sengstaken-Blakemore was first introduced? I believe our focus and interest on the perceptions of healthcare providers is a relatively modern one and perhaps it's unfair for us to measure a relatively old intervention by it.

These are all relevant points. We were surprised to find such a limited number of studies – largely conducted by the same research team. Of course this is a finding in itself and points to further research in this area. With regard to widening the searches we used 14 different search words and abbreviations (including ‘Sengstaken-Blakemore’) to identify relevant studies so we felt the UBT search terms were reasonably comprehensive. There may be some additional qualitative data ‘buried’ in survey type articles or overtly quantitative studies but these probably wouldn’t show up in qualitative searches. 

-While utilising Google Translate for a cursory search is efficient and practical, it may be advisable, given the epidemiology of PPH to cast a wider net and conduct targeted searches (ideally by RAs proficient in)in other languages such as French, Arabic etc

Fair point. We did try to incorporate a broad mix of countries and cultures by using continent specific databases (AJOL and LATINDEX) to identify studies from Africa and South America respectively. We didn’t exclude studies published in languages other than English and were ready to involve bilingual and trilingual researchers at the WHO to translate studies published in French, Spanish, Portuguese and Japanese as necessary. However, we just didn’t find many studies so didn’t utilize these resources. Arguably, we could have used language specific search terms but, as with many systematic reviews, we were working to a deadline and this would have slowed our progress considerably. Nevertheless we have added a comment in the discussion section (under limitations) to highlight that language specific search terms may yield more results. 

-Line 90: typo in Sengstaken-Blake(r)more

Thank you. This has been amended

-Line 133: Is "nurs" a typo?

‘Nurs*’ is an abbreviated search word which is used to identify studies including the terms ‘nurses’ and ‘nursing’. 

We hope these amendments are agreeable to the editors and the reviewers and look forward to hearing from you soon.

---

## [Decision Letter · Decision Letter 1]

3 Mar 2021

Healthcare providers experiences of using Uterine Balloon Tamponade (UBT) devices for the treatment of post-partum haemorrhage (PPH): a meta-synthesis of qualitative studies

PONE-D-20-30548R1

Dear Dr. Finlayson,

We’re pleased to inform you that your manuscript has been judged scientifically suitable for publication and will be formally accepted for publication once it meets all outstanding technical requirements.

Kind regards,

Simone Garzon

Academic Editor

PLOS ONE

Additional Editor Comments (optional):

Reviewers' comments:

Reviewer's Responses to Questions

**Comments to the Author**

1. If the authors have adequately addressed your comments raised in a previous round of review and you feel that this manuscript is now acceptable for publication, you may indicate that here to bypass the “Comments to the Author” section, enter your conflict of interest statement in the “Confidential to Editor” section, and submit your "Accept" recommendation.

Reviewer #1: All comments have been addressed

Reviewer #2: All comments have been addressed

Reviewer #3: All comments have been addressed

2. Is the manuscript technically sound, and do the data support the conclusions?

Reviewer #1: Yes

Reviewer #2: Yes

Reviewer #3: Yes

3. Has the statistical analysis been performed appropriately and rigorously? 

Reviewer #1: N/A

Reviewer #2: N/A

Reviewer #3: Yes

4. Have the authors made all data underlying the findings in their manuscript fully available?

Reviewer #1: Yes

Reviewer #2: Yes

Reviewer #3: Yes

5. Is the manuscript presented in an intelligible fashion and written in standard English?

Reviewer #1: Yes

Reviewer #2: Yes

Reviewer #3: Yes

6. Review Comments to the Author

Reviewer #1: I have no further comments. This is very well written and will greatly help the field as we all move toward the SDG targets

Reviewer #2: Thank you for the additional opportunity to review the manuscript. The issues presented in the review were addressed appropriately and I recommend this manuscript for publication.

Reviewer #3: Thank you for taking all the comments the various reviewers have made into consideration and providing an amended version. In as much as I agree that further studies are needed to properly elucidate the efficacy as well as user experiences (provider and patient) with UBT, it is my hope that even more resources (funding, technical know-how and research interest) will be channelled towards the social determinants that make the largest impact in reducing maternal mortality

7. PLOS authors have the option to publish the peer review history of their article (what does this mean?). If published, this will include your full peer review and any attached files.

Reviewer #1: **Yes: **Thomas F Burke, MD

Reviewer #2: **Yes: **Katherine J Kissler

Reviewer #3: **Yes: **Dr. Sandra Danso-Bamfo

---

## [Editor Report · Acceptance letter]

10 Mar 2021

PONE-D-20-30548R1 

Healthcare providers experiences of using Uterine Balloon Tamponade (UBT) devices for the treatment of post-partum haemorrhage: a meta-synthesis of qualitative studies 

Dear Dr. Finlayson:

I'm pleased to inform you that your manuscript has been deemed suitable for publication in PLOS ONE. Congratulations! Your manuscript is now with our production department. 

Kind regards, 

on behalf of

Dr. Simone Garzon 

Academic Editor

PLOS ONE